# Production of Astaxanthin by Animal Cells via Introduction of an Entire Astaxanthin Biosynthetic Pathway

**DOI:** 10.3390/bioengineering10091073

**Published:** 2023-09-11

**Authors:** Yousef Mohammed, Ding Ye, Mudan He, Houpeng Wang, Zuoyan Zhu, Yonghua Sun

**Affiliations:** 1State Key Laboratory of Freshwater Ecology and Biotechnology, Key Laboratory of Breeding Biotechnology and Sustainable Aquaculture, Institute of Hydrobiology, Chinese Academy of Sciences, Wuhan 430072, China; ymy.200012@gmail.com (Y.M.); yeding@ihb.ac.cn (D.Y.); hemudan@ihb.ac.cn (M.H.); wanghp@ihb.ac.cn (H.W.); zyzhu@ihb.ac.cn (Z.Z.); 2College of Advanced Agricultural Sciences, University of Chinese Academy of Sciences, Beijing 100049, China; 3Hubei Hongshan Laboratory, Wuhan 430072, China

**Keywords:** astaxanthin, metabolic engineering, animal cells in vitro, ketocarotenoids, pathway, biosynthesis, enzymatic reactions, multicistronic vectors

## Abstract

Astaxanthin is a fascinating molecule with powerful antioxidant activity, synthesized exclusively by specific microorganisms and higher plants. To expand astaxanthin production, numerous studies have employed metabolic engineering to introduce and optimize astaxanthin biosynthetic pathways in microorganisms and plant hosts. Here, we report the metabolic engineering of animal cells in vitro to biosynthesize astaxanthin. This was accomplished through a two-step study to introduce the entire astaxanthin pathway into human embryonic kidney cells (HEK293T). First, we introduced the astaxanthin biosynthesis sub-pathway (Ast subp) using several genes encoding β-carotene ketolase and β-carotene hydroxylase enzymes to synthesize astaxanthin directly from β-carotene. Next, we introduced a β-carotene biosynthesis sub-pathway (β-Car subp) with selected genes involved in Ast subp to synthesize astaxanthin from geranylgeranyl diphosphate (GGPP). As a result, we unprecedentedly enabled HEK293T cells to biosynthesize free astaxanthin from GGPP with a concentration of 41.86 µg/g dry weight (DW), which represented 66.19% of the total ketocarotenoids (63.24 µg/g DW). Through optimization steps using critical factors in the astaxanthin biosynthetic process, a remarkable 4.14-fold increase in total ketocarotenoids (262.10 µg/g DW) was achieved, with astaxanthin constituting over 88.82%. This pioneering study holds significant implications for transgenic animals, potentially revolutionizing the global demand for astaxanthin, particularly within the aquaculture sector.

## 1. Introduction

Astaxanthin (3,3′-Dihydroxy-β, β-carotene-4,4′-dione) is a high-value molecule classified as a ketocarotenoid, synthesized through a complex pathway, and commonly found as a unique red-orange pigment in many animals such as salmon, trout, flamingos, and crustaceans (shrimp, crab, krill, and crayfish) [1,2]. Although all animals cannot synthesize carotenoids de novo, except aphids (*Aphis* spp.) [3,4], they accumulate it from zooplankton, which consumes astaxanthin from phytoplankton [5]. Astaxanthin shares a similar structure to β-carotene: a conjugated double bond chain (polyene chain) and two terminal carbon rings (ionone ring). However, it differs from β-carotene by having a ketone and a hydroxyl group on each ring [6,7]. This structure explains several features of astaxanthin, including its superior antioxidant properties and its ability to be esterified. However, due to its high lipophilicity and low bioavailability, it is difficult to evaluate its efficacy in animal models for human pathologies and to estimate its health benefits. Despite this challenge, numerous studies have demonstrated that astaxanthin can protect against cancer, diabetes, cardiovascular disease (CVD), and other chronic conditions [8]. Therefore, astaxanthin has been widely used in nutritional supplements and in the cosmetics and pharmaceutical industries besides being employed as a feed supplement, mainly for salmon, trout, and shrimp aquaculture [9].

Synthetic astaxanthin dominates the world market due to the mass production cost of natural astaxanthin [1]. Recently, interest in natural sources of astaxanthin has increased considerably because of the safety issues with synthetic astaxanthin. Although natural astaxanthin is widely synthesized by *Haematococcus lacustris* (formerly called *Haematococcus pluvialis*) and *Xanthophyllomyces dendrorhous* [10,11], it is still limited, expensive, and only serves medical and pharmaceutical applications. Consequently, researchers have made numerous attempts to optimize astaxanthin production in natural producers, whereas the astaxanthin pathway has been introduced to heterologous hosts such as *Escherichia coli*, *Saccharomyces cerevisiae*, and *Bacillus megaterium* for large-scale production of sustainable natural astaxanthin [12,13]. In addition, the value of some plants, such as rice and soybean, were improved by introducing the astaxanthin biosynthetic pathway using metabolic engineering approaches [4,14,15,16].

Over the past two decades, metabolic engineering has emerged as a powerful tool for manipulating cellular systems to produce many valuable compounds in microorganisms and plants [17,18] by optimizing or introducing novel pathways [19,20,21]. Recently, metabolic engineering has enabled the use of in vitro and in vivo animals as sophisticated cell factories to manufacture a wide range of products for medical, pharmaceutical, and other biotechnological applications [22,23]. It has also contributed to increasing the value of some animals, where a natural biosynthetic pathway is transferred to gain a new trait or ability to produce high-value molecules such as polyunsaturated fatty acids [24,25].

In our previous study, we demonstrated the efficiency of the metabolic engineering approach to grant fish cells and embryos the ability to manufacture ketocarotenoids from β-carotene as a substrate. This was achieved by using *crtW* and *crtZ* synthesized genes, which encode the β-carotene ketolase and β-carotene hydroxylase enzymes [26]. Additionally, we utilized multicistronic expression vectors with self-cleaving 2A peptides (P2A) between genes to enable the simultaneous expression of multiple genes within animal cells. This method significantly increased the total content of ketocarotenoids compared to single-gene expression systems [26,27].

In this study, our major objective was to evaluate the potential of using in vitro animal cells as cell factories for astaxanthin biosynthesis from geranylgeranyl pyrophosphate (GGPP). To accomplish this, we employed recent techniques in metabolic engineering to introduce the entire astaxanthin pathway into human embryonic kidney 293T cells (HEK293T), a suitable in vitro animal cell model with high transfection efficiency and expression levels. HEK293T cells are ideal for both stable and transient gene transfer methods, as well as robustness and ease of maintenance [28,29]. This cell line, a variant of HEK293 cells that expresses Simian virus 40 (SV40) large T antigen, allows for the amplification of vectors containing the SV40 origin. This significantly increases exogenous protein expression levels during transient gene transfer [28,30].

Engineering the metabolic pathway of astaxanthin requires introducing multiple genes to the new host simultaneously; astaxanthin is biosynthesized by approximately five enzyme-mediated steps proceeding from GGPP towards astaxanthin [31,32]. To simplify and better understand the astaxanthin biosynthetic pathway, we divided the entire pathway into two sub-pathways: (i) the astaxanthin biosynthesis sub-pathway (Ast subp), and (ii) the β-carotene biosynthesis sub-pathway (β-Car subp), as illustrated in Figure 1A. Several genes/enzymes involved in the two sub-pathways of astaxanthin biosynthesis were investigated through a two-step study to determine appropriate genes/enzymes for metabolic engineering of HEK293T cells to manufacture astaxanthin from GGPP. In the first step of the study, we introduced Ast subp into HEK293 cells to synthesize astaxanthin directly from β-carotene. This occurred by constructing three multicistronic expression vectors to combine several artificially synthesized genes (their codon sequences were optimized for animals and artificially synthesized) encoding β-carotene ketolase (*crtZ* and *H.crtZ* genes) and β-carotene hydroxylase enzymes (*crtW*, *bkt3*, and *cbkI* genes). These vectors were then transfected into HEK293 cells to determine the best-characterized genes/enzymes for astaxanthin biosynthesis as a final product. We then introduced two novel genes (*DGTT1* and *DGTT2*), which encode diacylglycerol acyltransferase enzymes, combined with the best-characterized genes from Ast subp to modify astaxanthin molecules to be monoesters or diesters and increase astaxanthin stability [32,33]. In the second step, we redirected GGPP flux towards astaxanthin accumulation in HEK293T cells by introducing β-Car subp within three multicistronic expression vectors. These three vectors included several artificially synthesized genes that encode phytoene synthase (*psy1* gene), phytoene desaturase *(Pa-crtI*, *Xd-crtI*, and *crtYB* genes), and lycopene β-cyclase (*lcyb* gene), along with the selected genes from Ast subp, to empower HEK293T cells to manufacture astaxanthin from GGPP. We subsequently optimized astaxanthin biosynthesis in animal cells in vitro by using critical factors in the astaxanthin biosynthetic process. Overall, our study represents a significant advance in the field of metabolic engineering by demonstrating the success of astaxanthin biosynthesis in animal cells through the introduction of the entire astaxanthin biosynthetic pathway into HEK293T cells. Our results highlight the enormous potential of metabolic engineering to revolutionize the production of valuable molecules in animal cells in vitro. With the increasing demand for natural astaxanthin, our findings have far-reaching implications for the aquaculture industry and open up exciting possibilities for the production of other high-value compounds using similar approaches.

## 2. Materials and Methods

### 2.1. Cell Culture and Transfection

Human embryonic kidney (HEK293T) cells were maintained in Dulbecco’s modified Eagle’s medium (DMEM (high glucose, Biological Industries, 01-052-1ACS)) supplemented with 10% (*v*/*v*) fetal bovine serum (FBS (Biological Industries, 04-001-1A)) at 37 °C in a humidified atmosphere with 5% CO_2_ as previously described [34,35]. HEK293T cells were transfected using VigoFect (Vigorous Biotechnology, China) according to the manufacturer’s instructions at a dosage of ten μg of plasmid for cells covering about 70% of the surface of the culture dish (Nest, 100 mm cell culture dish). After 8 h of incubation, the transfection complex medium was replaced with fresh medium supplemented with β-carotene (20-μg β-carotene per 1 mL medium), using tetrahydrofuran as a miscible co-solvent, or supplied with GGPP (2-μg GGPP per 10 mL medium). Sixteen hours post-transfection, transfected cells were imaged using a fluorescent microscope to evaluate transfection. After 72 h of incubation, the transfected cells, including control samples, were collected for ketocarotenoid extraction and analysis. All plasmids were transfected into HEK293T cells in parallel with three replicates, along with controls supplied with β-carotene or GGPP but without parameter studies.

#### Assessment of Transfection Effectiveness

A fluorescence microscopy procedure was employed to evaluate the transfection process’s efficacy. Sixteen hours post-transfection, all transfected cells were imaged using the Leica AF6000 Fluorescence Image System (Leica Microsystems, Wetzlar, Germany), controlled through the Leica AF6000 Software. This allowed us to observe the expression of target genes, including the enhanced green fluorescent protein (EGFP) and the red fluorescent protein (mCherry), indicating the co-expression efficiency of all genes. This is because all of the astaxanthin biosynthetic genes and the EGFP or mCherry reporter genes were under the same CMV promoter within an open reading frame (ORF) [26,36,37]. Using the fluorescent microscope with a 10× objective, we captured both bright field and fluorescent images to elucidate the extent of transfection. Specifically, a mercury-vapor lamp was used as the light source, and for EGFP, fluorescence was viewed through the GFP channel with blue light (BP 470/40) for excitation and green light (BP 525/50) for emission. Similarly, for mCherry, fluorescence observation occurred through the Y3 channel with green light (BP 535/50) for excitation and a wider range (BP 610/75) for emission. Subsequent quantification of transfection efficiency was achieved by deriving the ratio of the fluorescent cell number to the total cell number in a defined area by utilizing the Fiji software (v2.3.0.) [38], an approach applied separately for each fluorescence in the case of double transfection [24].

### 2.2. Synthetic Heterologous Genes Expression and Plasmid Construction

All DNA manipulations were carried out with standard cloning procedures [39], using restriction enzyme digestion and Gibson assembly methods [40]. All heterologous genes in this study (*crtW*, *crtZ* genes (GenBank: AB181388.1) from *Brevundimonas* sp.; *cbkI* gene (RefSeq: XM_001698647.1) from *Chlamydomonas reinhardtii*; *bkt3*, *H-crtZ lcyb, DGTT1,* and *DGTT2* genes (GenBank: AY603347.1, AY187011.1, KX424526.1, MN561785.1, MN561786.1) from *Haematococcus lacustris; psy1* gene (GenBank: FJ971252.1) originated from *Zea mays* subsp; *Pa-crtI* gene (GenBank: D90087.2) from *Pantoea ananatis*; and *Xd-crtI* and *crtYB* (GenBank: KR779666.1, AY177204.1) from *Xanthophyllomyces dendrorhous*) were codon-optimized for animals and artificially synthesized by TsingKe Biological Technology and Tianyi Huiyuan Bioscience & Technology Companies (Wuhan, China). 

In this study, we used the Tol2 transposon-based expression plasmid (pTol2-CMV-SV40 poly(A)) containing an individual cassette with one CMV promoter, as previously described [24,26]. To construct Ast subp plasmids (pWZG-2A, pbK3HZG-2A, and pCIHZG-2A), the genes *crtW* and *crtZ,* which originated from *Brevundimonas* sp., were subcloned into *ClaI/HindIII* and *HindIII/EcoRI* sites of the pTol2-CMV-SV40 poly(A) vector, respectively. The *EGFP* gene was inserted at the *EcoRI/XbaI* site of the pTol2-CMV-SV40 poly(A) vector downstream of the *crtZ* genes in order to evaluate the expression efficiency. The self-cleaving 2A peptides P2A and T2A were used between genes to simultaneously express multiple genes with relatively high levels of downstream protein expression. The 2A peptides P2A and T2A were inserted directly upstream of the *crtZ* gene and *EGFP*, respectively, to yield the pWZG-2A construct (Figure 1D). To construct plasmid pbK3HZG-2A, the *sbkt3* and *H.crtZ* genes from *H. lacustris* were relocated at *ClaI/HindIII* and *HindIII/EcoRI* sites of the pTol2-CMV-2A-*EGFP*-SV40 poly (A) (or pG-2A) expression vector, respectively. The 2A peptide (P2A) was added upstream of the *H.crtZ* gene (Figure 1F). The gene *cbkI* from *C. reinhardtii* was relocated at the *ClaI*/*HindIII* site of the pTol2-CMV-HZ-EGFP-2A-SV40 poly(A) (or pHZG-2A) construct to generate the pCIHZG-2A plasmid (Figure 1E). The pWZg1R-2A and pWZg2R-2A expression plasmids were constructed to evaluate the efficiency of the *DGTT1* and *DGTT2* genes within the astaxanthin biosynthetic pathway. The *mCherry* gene was first relocated to the *EcoRI/XbaI* site of the pWZG-2A plasmid instead of *EGFP* and released as pWZR-2A. We then subcloned the *DGTT1* and *DGTT2* genes into *SpeI/XhoI* of the pWZR-2A construct separately to generate pWZg1R-2A and pWZg2R-2A plasmids, respectively, considering the insertion of the 2A peptides (E2A) upstream of the *mCherry* gene (Figure 1B,C).

For the construction of β-Car subp plasmids, the artificially synthesized genes, *psy1* gene from *Zea mays* subsp, *lcyb* gene from *H. lacustris*, and *Xd-crtI* gene from *X. dendrorhous,* were introduced to the pTol2-CMV-EGFP-2A-SV40 poly(A) (or pG-2A) plasmid at *ClaI*/*HindIII*, *HindIII*/*EcoRI*, and *EcoRI*/*BamHI* sites, respectively. The 2A peptides P2A, T2A, and E2A were inserted upstream of the *lcyb*, *Xd.crtI*, and *EGFP* genes consecutively to construct the pPLXG plasmid (Figure 1H). In order to construct the pPYLG plasmid, the *crtYB* from *X. dendrorhous* and the *lcyb* gene were relocated into the *HindIII*/*EcoRI* and *EcoRI*/*BamHI* sites of the pPLXG plasmid instead of the *lcyb* and *Xd.crtI* genes (Figure 1G). The genes *Pa-crtI* from *P. ananatis* and *psy1* were relocated in place of psy1 and *crtYB* genes at the *ClaI*/*HindIII* and *HindIII*/*EcoRI* sites of the pPYLG plasmid to yield the pAPLG expression plasmid (Figure 1I).

The sequence encoding of all 2A peptides, P2A, T2A, and E2A, were synthesized by extension PCR using appropriate primers. The nucleotide sequences encoding Gly-Ser-Gly (GSG) were added to the N-terminus of P2A, T2A, and E2A to improve cleavage efficiency [26,37]. A 9 bp Kozak sequence GCCGCCACC was added directly before the ATG start codon of the first gene in all above-mentioned constructs as the optimum sequence for initiating translation. In addition, the stop codon TAA was added only at the end of the last gene in every construct mentioned above to generate an open reading frame (ORF) [27]. 

Finally, the plasmid pcrtZ-R was constructed by subcloning the genes *crtZ* and *mCherry* gene into *HindIII/EcoRI* and *ClaI/SpeI* of the plasmid pTol2-CMV-SV40 poly(A)-CMV-SV40 poly(A), which contains two separate expression cassettes with two CMV promoters for each one (dual gene expression cassettes). All artificially synthesized genes were amplified through PCR using specific corresponding primers as listed in Appendix A. 

### 2.3. Ketocarotenoid Extraction 

Extraction of ketocarotenoids was conducted following the protocol outlined in a previous study [41]. Briefly, HEK293T cell samples were collected by centrifuging the culture medium at 3000 rpm for 10 min at 4 °C following cell dissociation using trypsin. The supernatant was discarded, and the cells were washed twice with phosphate-buffered saline (PBS) before being dried by an oven (50–55 °C) to determine their dry weight (DW). Subsequently, the HEK293T cells were homogenized in a 10-mL glass tube containing 2 mL of an extraction solvent consisting of 25% (*v*/*v*) dichloromethane and 75% (*v*/*v*) methanol, with a ratio of 1:3 (*v*/*v*) of dichloromethane to methanol. For cell disruption, ultrasonication (Sonics & Materials, Inc., 53 Church Hill Road, Newtown, CT 06470-1614, USA) was performed five times, with 1 min intervals under ice. Following this step, centrifugation at 4000 rpm for 5 min at 4 °C separated the cell extraction solution from the disrupted cells. The resulting supernatant, enriched with carotenoids, was carefully collected into a fresh 10 mL glass tube. This process was repeated three times. Subsequently, the combined extraction mixtures were centrifuged again (4000 rpm) for 5 min at 4 °C, and the supernatant was then transferred to a fresh 10 mL glass tube. The extraction mixture was subsequently evaporated under a stream of nitrogen to remove the solvent. After evaporation, all samples were dissolved in 0.5 mL of the extraction solvent, filtered, and finally stored in the dark at −80 °C for subsequent analysis.

### 2.4. Ketocarotenoid Analysis

Ketocarotenoids were purified and analysed using reversed-phase high-performance liquid chromatography (RP-HPLC), following a previously established method [41]. In brief, a Waters liquid chromatography system equipped with a 996-photodiode array detector (Waters, Milford, MA, USA) was utilized. The separation of all carotenoids was conducted using a Waters Symmetry C18 column (5 µm; 150 × 4.6 mm) at room temperature. The mobile phase consisted of two constituents: eluent A (composed of dichloromethane: methanol: acetonitrile: water, 5.0:85.0:5.5:4.5, *v*/*v*) and eluent B (composed of dichloromethane: methanol: acetonitrile: water, 25.0:28.0:42.5:4.5, *v*/*v*). The separation of carotenoids followed this gradient protocol, initiating with an 8 min elution with 0% of eluent B. Subsequently, a linear gradient to 100% eluent B was executed over 6 min, followed by a continuous run of 100% eluent B for 40 min. The flow rate was consistently maintained at 1.0 mL/min. For carotenoid detection, absorption spectra were displayed within the range of 250 to 700 nm, with specific measurements taken at 480 nm. The identification of all ketocarotenoids relied on both ketocarotenoid standards and the distinctive UV absorption profiles of carotenoids (Appendix A) [5]. GGPP and all carotenoid standards, including β-carotene, canthaxanthin, and astaxanthin, were procured from Sigma-Aldrich (St. Louis, MO, USA). The presented data are depicted as means ± standard deviation (SD) for three biological replicates.

### 2.5. Statistical Analysis

Origin 2019 software (OriginLab Corp., Northampton, MA, USA) was used in this study. Mean values along with their corresponding standard deviations (SD) were calculated from the data collected in three separate replicates. A one-way ANOVA was employed to determine the significant variations between diverse groups in our single-factor experiments. *p*-values were calculated and represented as *p* < 0.05, *p* < 0.01, and *p* < 0.001 and were indicated by *, **, and ***, respectively. Appendix A illustrate the statistical tests of this study and all the *p*-values.

### 2.6. Nucleotide Accession Number

The nucleotide sequences of all artificially synthesized genes mentioned in this study can be found in the GenBank data libraries (https://www.ncbi.nlm.nih.gov/genbank/, accessed on 15 April 2023) under accession numbers: MT360309 (*crtW*), MT360311 (*crtZ*), OR402794 (*bkt3*), OR402795 (*H-crtZ*), OR402796 (*cbkI*), OR402797 (*DGTT1*), OR402798 (*DGTT2*), OR402799 (*psy1*), OR402800 (*lcyb*), OR402801 (*Pa-crtI*), OR402802 (*Xd-crtI*), and OR402803 (*crtYB*).

## 3. Results and Discussion

### 3.1. Genes/Enzymes Selection for Astaxanthin Biosynthesis in Animal Cells In Vitro

In order to engineer the metabolic pathway of astaxanthin in animal cells in vitro, we were required to introduce various genes to the HEK293T cells simultaneously. Throughout this study, we investigated several genes/enzymes involved in astaxanthin pathways for the genetic manipulation of HEK293T cells to manufacture astaxanthin from GGPP. This was accomplished through a two-step study to introduce multiple artificially synthesized genes encoding the entire astaxanthin pathways, including Ast subp and β-Car subp.

In Ast subp, β-carotene undergoes a series of reactions to add two hydroxyl and keto moieties to each β-ionone ring to accumulate astaxanthin as a final product via hydroxylase and ketolase enzymes [7]. Our previous study has identified the best-characterized genes encoding these enzymes in the marine bacteria *Brevundimonas* sp. (*crtW* and *crtZ*) [26]. In this study, we expand on our previous research by exploring additional astaxanthin biosynthesis genes derived from the green algae *H. lacustris* (*bkt3* and *H.crtZ*) and *C. reinhardtii* (*cbkI*).

In the β-Car subp, two molecules of GGPP are combined to form phytoene via phytoene synthase. The phytoene desaturase inserts four double bonds in phytoene to obtain lycopene. After desaturation, lycopene cyclase converts the ψ acyclic ends of lycopene as β-ionone to form β-carotene [6,7,31]. Numerous studies have identified the most effective genes encoding these enzymes from various sources: *psy1* gene encoding phytoene synthase from *Zea mays* subsp, *Pa-crtI* from *Pantoea ananatis*, *Xd-crtI* and *crtYB* from *Xanthophyllomyces dendrorhous* for phytoene desaturase, and *lcyb* gene encoding lycopene β-cyclase from *H. lacustris* [14,15]. Thus, these sets of genes were selected for our second pilot study to redirect GGPP flux from the animal mevalonate pathway to astaxanthin biosynthesis in HEK293T cells. All of these genes were codon-optimized for animals and artificially synthesized to be expressed in HEK293T cells.

### 3.2. Constructing Ast subp in Animal Cells In Vitro 

To enable animal cells to biosynthesize astaxanthin from β-carotene in vitro, three multicistronic expression vectors, pWZG-2A, pbK3HZG-2A, and pCIHZG-2A, were transfected into HEK293T cells. As shown in Figure 2, approximately 45–65% of the cells transfected with pWZG-2A, pbK3HZG-2A, and pCIHZG-2A plasmids showed strong expression of green fluorescent protein, which indicated the high efficiency of the co-expression of all genes since all of the genes, including the *EGFP* gene, in the three plasmids (pWZG-2A, pbK3HZG-2A, and pCIHZG-2A) were under one CMV promoter within an ORF. 

According to the RP-HPLC results (Figure 2A–C), we successfully empowered the animal cells in vitro to biosynthesize astaxanthin from β-carotene, as compared with non-transfected cells supplied with β-carotene as a control. Astaxanthin and canthaxanthin with their *E* (*trans*) and Z (*cis*) isomers were present in the cell extraction of transfected HEK293T cells by pWZG-2A plasmids with a total ketocarotenoid concentration of 74.21 µg per gram of DW as illustrated in Figure 2A. The concentration was significantly higher than the HEK293T cell extracts transfected with pCIHZG-2A and pbK3HZG-2A plasmids, which released ketocarotenoids with concentrations of 55.61 and 40.24 µg/g DW, respectively (Figure 2B,C). The total astaxanthin concentration was likewise significantly higher in the cell-extracts transfected with pWZG-2A plasmids (31.59 µg/g DW) compared with cell extracts transfected with pCIHZG-2A and pbK3HZG-2A plasmids (Figure 2F and Table 1). 

Based on these findings and consistent with our previous study [26], the artificially synthesized genes (*crtW* and *crtZ*) derived from the marine bacteria *Brevundimonas* sp. were demonstrated to be the best selection genes for genetic manipulation inside in vitro animal cells to biosynthesize astaxanthin from β-carotene, compared with the artificially synthesized genes *bkt3*, *cbkI,* and *H.crtZ*, which originated from the green algae *H. lacustris* and *C. reinhardtii*. Therefore, these genes (*crtW* and *crtZ*) were considered for further study to increase the astaxanthin content and stability in HEK293T cells. Furthermore, the exploration of alternate genes sourced from various microorganisms, such as *Pseudospongiococcum protococcoides*, *Haematococcus rubicundus*, and *Coelastrella aeroterrestrica* warrant in-depth examination in future research endeavors [42].

### 3.3. Introduction of DGTT1 and DGTT2 Genes for Optimization of Astaxanthin Production and Stability

As reported in several studies, it has been proposed that the diacylglycerol acyltransferases enzyme, which is encoded by the *DGTT1* and *DGTT2* genes, may catalyze astaxanthin esterification in several green algae to accumulate astaxanthin monoesters or diesters, which are more stable than free astaxanthin [32,33]. In addition, this enzyme likely contributes to the efficient deposition of carotenoids within the cell’s hydrophobic compartments [42]. Therefore, the pWZg1R-2A and pWZg2R-2A plasmids were transfected into HEK293T cells separately with control and supplied with β-carotene to investigate their potential for enhancing astaxanthin production and stability. After 16 h of transfection, all specimens were imaged by fluorescence microscopy. We found that approximately 50–75% of the cells transfected with pWZg1R-2A and pWZg2R-2A plasmids revealed a high expression of mCherry fluorescent proteins, which implies the high efficiency of plasmid transfection and the high expression levels of all genes included (Figure 2D,E).

Based on RP-HPLC data, as shown in Figure 2D,E, astaxanthin ester was not detected in cell extracts that were transfected with pWZg1R-2A and pWZg2R-2A plasmids; instead, canthaxanthin was the dominant ketocarotenoid with a low content of free astaxanthin. Consequently, neither *DGTT1* nor *DGTT2* genes are able to catalyze astaxanthin esterification in HEK293T cells. However, the introduction of the *DGTT1* gene combined with astaxanthin biosynthesis genes (*crtW* and *crtZ*) promoted the accumulation of ketocarotenoids, particularly in canthaxanthin, which increased 2.49 times (106.1 µg/g DW) compared with transfection with only *crtW* and *crtZ* genes (42.62 µg/g DW), as illustrated in Figure 2F and Table 1. This enhancement is likely attributed to the role of diacylglycerol acyltransferases enzymes, which appear to play a significant part in depositing ketocarotenoids within the hydrophobic compartments of the cell. Thus, the pWZg1R-2A expression plasmid, which includes the *crtW, crtZ, and DGTT1* genes, was further used in the next step of our study to biosynthesize astaxanthin from GGPP.

### 3.4. Redirection of GGPP Flux towards Astaxanthin Accumulation

GGPP is a prenyl diphosphate and represents a key precursor for astaxanthin biosynthesis. In animal cells, GGPP is involved in the post-translational modification process of proteins by the covalent attachment of an isoprenoid lipid (prenylation), which is necessary for animal growth, differentiation, and morphology [43,44]. GGPP is biosynthesized exclusively through a series of enzymatic reactions within the mevalonate biosynthetic pathway from 3-hydroxy-3-methylglutaryl-CoA (HMG-CoA), as shown in Appendix A. Unlike microorganisms and higher plants which can synthesize GGPP through the non-mevalonate pathway [44,45].

In this part of the study, we redirected GGPP flux from the animal mevalonate pathway towards astaxanthin accumulation in HEK293T cells. This was achieved by introducing the β-Car subp within the pPYLG, pPLXG, and pAPLG constructed plasmids to HEK293T cells along with the selected genes from the Ast subp (pWZg1R-2A plasmid). As shown in Figure 3A, nearly 45–65% of the cells transfected with pPYLG, pPLXG, and pAPLG plasmids along with the pWZg1R-2A plasmid showed strong expression of green and mCherry fluorescent proteins, which indicated the high efficiency of co-expression of all genes inserted upstream of *EGFP* and *mCherry* genes.

As confirmed by RP-HPLC data (Figure 3B), we successfully induced HEK293T cells as an in vitro animal model for the first time to biosynthesize astaxanthin from the metabolic flux of GGPP by introducing β-Car subp, as compared with the control sample. Astaxanthin was the major ketocarotenoid, besides a low content of canthaxanthin in the cell extracts of the transfected cells. The highest concentration of total ketocarotenoids was in the cell extracts transfected by pPYLG and pWZg1R-2A plasmids with 63.24 µg/g DW (Figure 3C,D), which is significantly higher, compared with the cells transfected with pPLXG and pAPLG plasmids along with the pWZg1R-2A plasmid (43.48 and 51.74 µg/g DW, respectively), as listed in Table 1. Although the concentration of total ketocarotenoids was relatively low in the transfected cell with pPYLG and pWZg1R-2A plasmids, the quality of extracted ketocarotenoids was comparatively high due to the high percentage of astaxanthin (66.19%), as shown in the pie charts in Figure 3E.

From these results, it is obvious that introducing β-Car subp along with Ast subp to HEK293T cells redirected the GGPP flux towards astaxanthin formation. However, the concentration of ketocarotenoids was relatively low. This could be due to low GGPP synthesis or leakage through other pathways, leading to a weak flow rate of GGPP through the mevalonate pathway. Therefore, further investigation was required to enhance the GGPP flux towards astaxanthin pathways, thereby increasing astaxanthin accumulation.

### 3.5. Optimization of the Astaxanthin Biosynthetic Pathway through Exogenous GGPP

In order to increase the GGPP flux towards astaxanthin accumulation in animal cells in vitro, we introduced pPYLG, pPLXG, and pAPLG expression plasmids to HEK293T cells combined with the pWZg1R-2A plasmid and supplied with GGPP (2-μL GGPP/10 mL medium). After 24 h of incubation, we observed that approximately 55–75% of the cells transfected with pPYLG, pPLXG, and pAPLG plasmids along with the pWZg1R-2A plasmid showed high expression of green and mCherry fluorescent proteins (Figure 4A) providing evidence for the successful co-expression of all genes upstream of the *EGFP* and *mCherry* genes.

As substantiated by RP-HPLC data (Figure 4B and Figure 5C), we significantly optimized the astaxanthin biosynthetic pathway in HEK293T cells. The total ketocarotenoid concentration remarkably increased up to 3.28-fold (207.19 µg/g DW) in the transfected cell extracts with pPYLG and pWZg1R-2A plasmids and supplied with exogenous GGPP, compared with transfected cell extracts with the same plasmids and without GGPP (63.24 µg/g DW) (Figure 4C and Figure 5F), which turned the cells to a red hue, as shown in Figure 4E, due to ketocarotenoid accumulation. The total ketocarotenoid content in the transfected cell extracts with the pAPLG and pPLXG plasmids combined with the *pWZg1R*-2A plasmid was significantly increased as well, up to 2.4- and 1.74-fold (124.41 and 75.87 µg/g DW), respectively, when cells were supplied with exogenous GGPP (Figure 4D and Table 1).

Although the total ketocarotenoid concentration was remarkably increased by supplying the transfected cells with GGPP, the astaxanthin content was relatively low. As shown in Figure 4D and the pie charts in Figure 4F, the astaxanthin ratio dropped obviously from 66.19% to 26.1% of total ketocarotenoids. It is evident that the significant increase in total ketocarotenoid concentration was due to the high content of canthaxanthin accumulation. Canthaxanthin is synthesized from β-carotene through two reactions using only the β-carotene ketolase enzyme encoded by the *crtW* gene, whereas astaxanthin synthesis requires at least four intermediate reactions catalyzed by two enzymes, β-carotene ketolase (crtW) and β-carotene hydroxylase (crtZ). Consequently, the high concentration of canthaxanthin might be explained by the high efficiency of the β-carotene ketolase enzyme (crtW). In contrast, the low content of astaxanthin might be due to the low efficiency of β-carotene hydroxylase (crtZ). It is likely that the incorporation of 2A peptide for the construction of multicistronic expression vectors may have slightly altered the conformation of β-carotene hydroxylase (crtZ) near the P2A region, where it is predicted to be the first helix of crtZ involved in the active site, as shown in Appendix A [13]. Thus, this change could disrupt the correct functioning of the crtZ enzyme. Therefore, a further optimization step was required to improve the quality of ketocarotenoid production.

### 3.6. Qualitative Optimization of Total Ketocarotenoids by Individual Expression of the crtZ Gene

In an effort to enhance the quality of ketocarotenoids synthesized in HEK293T cells in terms of carotenoid composition by increasing the concentration of the target product (astaxanthin), we introduced the *crtZ* gen (encoding the β-carotene hydroxylase enzyme) to HEK293T cells within the pWZg1R-2A plasmid, an individual gene expression vector, to avoid inhibiting the function of the enzyme. This plasmid was transfected with pPYLG, pPLXG, and pAPLG plasmids separately into HEK293T cells, combined with the pWZg1R-2A plasmid, and supplied by GGPP. After 24 h of incubation, transfected HEK293T cells were imaged by fluorescence microscope, and we found that 60–75% of the transfected cells exhibited a robust expression of green and mCherry fluorescent proteins, as shown in Figure 6A. This demonstrated the effective expression of all genes involved in transfected plasmids.

RP-HPLC data (Figure 5E and Figure 6B) confirmed a substantial improvement in the quality of ketocarotenoid composition through a significant increase in the astaxanthin content by 4.3-fold (232.81 µg/g DW), which represented 88.82% of the total ketocarotenoids (262.10 µg/g DW) in the transfected cell extracts with the pcrtZ-R plasmid compiled with pPYLG and pWZg1R-2A plasmids and supplied with GGPP, as compared to without the pcrtZ-R plasmid (Figure 5F and Figure 6C). This high astaxanthin content resulted in a distinct red coloration observed in transfected cells, as shown in Figure 6E.

The concentration of astaxanthin also increased substantially (up to 3.16- and 7.50-fold) when the pcrtZ-R plasmid was introduced with the two other trials (pAPLG+pWZg1R-2A plasmids and pPxLG+pWZg1R-2A plasmids) and provided with exogenous GGPP. Astaxanthin levels reached 136.74 and 146.39 µg/g DW, respectively, which accounted for 70–94% of the total ketocarotenoids, as illustrated in Figure 6D and the pie charts in Figure 6F.

These outstanding results supported our observations and are in line with previous studies on β-carotene hydroxylase (crtZ) enzymes [26]. The expression of crtZ within a multicistronic expression vector or fusion gene rather than individual expression may slightly modify the conformation of the β-carotene hydroxylase enzyme, preventing it from functioning properly.

## 4. Conclusions

Throughout this study, we successfully metabolically engineered HEK293T cells as an in vitro model for animal cells to synthesize astaxanthin from GGPP by introducing several artificially synthesized genes involved in the astaxanthin biosynthetic pathway using multicistronic expression vectors. Additionally, we significantly optimized the quantity and quality of ketocarotenoids synthesized in HEK293T cells through two optimization steps. These optimization efforts led to a 4.14-fold increase in the concentration of ketocarotenoids (262.10 µg/g DW), and astaxanthin content reached more than 88% of the total ketocarotenoids, as outlined in Figure 7. Our findings demonstrate the potential of in vitro animal cells as efficient platforms for astaxanthin biosynthesis. Although the metabolic engineering of in vitro animal cells for astaxanthin production has faced challenges such as limited yields, restricted substrate flux, competition with natural sources, and scalability complexities, the rapid advance in metabolic engineering proposes viable solutions, thereby paving the way for future studies aimed at developing transgenic animals with the ability to synthesize astaxanthin. This innovative strategy holds the potential to revolutionize global astaxanthin demand, especially within the aquaculture industry. Furthermore, engineered animal models producing astaxanthin provide a promising avenue to explore the potential health benefits of this potent antioxidant, despite its previously noted low bioavailability. Overall, this study establishes a foundation for future research into animal-based astaxanthin synthesis, highlighting the potential for pioneering advancements in metabolic engineering.

## Figures and Tables

**Figure 1 bioengineering-10-01073-f001:**
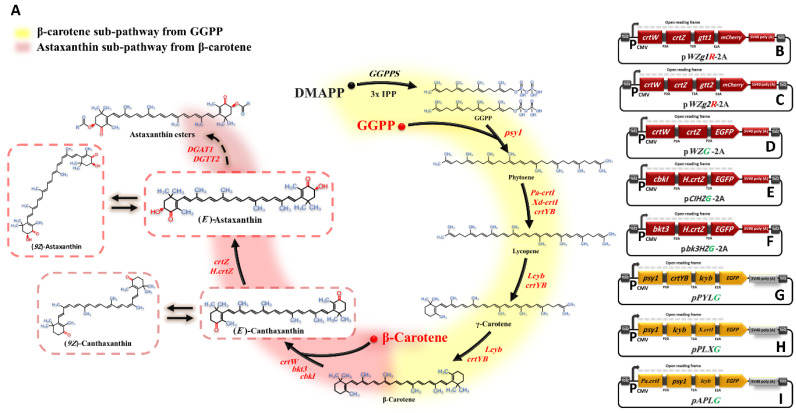
Astaxanthin biosynthesis pathway and constructed plasmids. (**A**) Simplification of the astaxanthin biosynthetic pathway reconstructed in HEK293T cells. Genes involved in this pathway are as follows: *psy1* gene, phytoene synthase; *crtI* and *crtYB* genes, phytoene desaturase; *lcyb* gene, lycopene β-cyclase; *crtZ* and *H.crtZ* genes, β-carotene hydroxylase; *crtW*, *bkt3*, and *cbkI* genes, β-carotene ketolase; and *DGTT1* and *DGTT2* genes, diacylglycerol acyltransferases. The yellow and red background colours indicate the sub-pathways of β-carotene and astaxanthin, respectively. Dashed arrows indicate that reaction was not successfully completed. The heterologous (exogenous) genes and molecules are shown in red. (**B**–**I**) illustrate the constructed plasmids utilized in this study.

**Figure 2 bioengineering-10-01073-f002:**
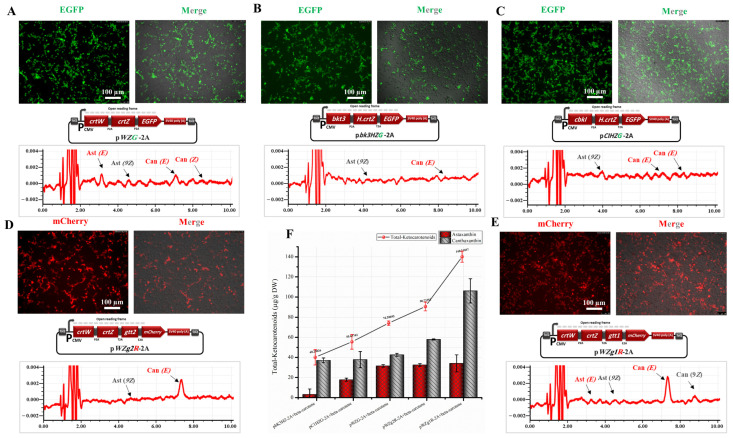
Assessment of the ability of animal cells (HEK293T cells) to synthesize astaxanthin in vitro from β-carotene. (**A**) Expression of EGFP and HPLC analysis of cells transfected with pWZG-A2 + β-carotene. (**B**) Expression of EGFP and HPLC analysis of cells transfected with pbk3HZG-A2 + β-carotene. (**C**) Expression of EGFP and HPLC analysis of cells transfected with pCIKHZG-A2 + β-carotene. (**D**) Expression of mCherry and HPLC analysis of cells transfected with pWZg2R-A2 + β-carotene. (**E**) Expression of mCherry and HPLC analysis of cells transfected with pWZg1R-A2 + β-carotene. Can, canthaxanthin; Ast, astaxanthin; *E*, *trans* geometric isomer; and *Z*, *cis* geometric isomer. (**F**) Ketocarotenoid concentration graph for HEK293T cell extracts transfected with the above-mentioned plasmids. DW, dry weight. Results were presented as means ± SD of three biological replicates.

**Figure 3 bioengineering-10-01073-f003:**
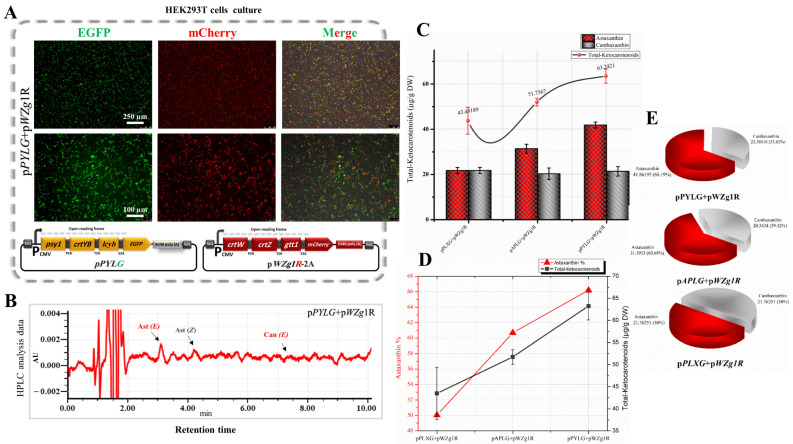
Construction of the astaxanthin biosynthetic pathway in HEK293T cells. (**A**) Expression of EGFP and mCherry in HEK293T cells transfected with pPYLG + pWZG-P2A plasmids. (**B**) HPLC analysis of cells transfected with above-mentioned plasmids. Can, canthaxanthin; Ast, astaxanthin; *E, trans* geometric isomer; and *Z, cis* geometric isomer. (**C**) Total ketocarotenoid concentration graph compared with the ratio of astaxanthin and canthaxanthin content. DW, dry weight. (**D**) Astaxanthin and canthaxanthin concentration graphs from cell extracts transfected with pPYLG + pWZG-P2A plasmids. (**E**) Pie charts show total ketocarotenoids with different percentages of astaxanthin and canthaxanthin content.

**Figure 4 bioengineering-10-01073-f004:**
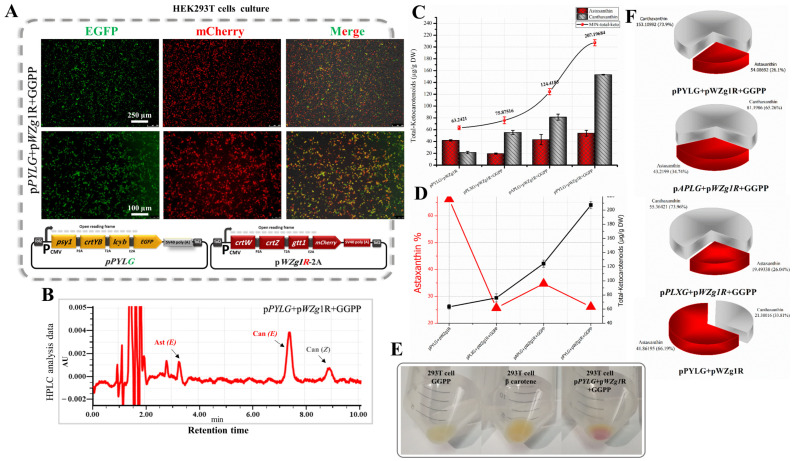
Evaluation of exogenous GGPP addition for total ketocarotenoid optimization in transfected HEK293T cells with β-carotene and astaxanthin biosynthetic genes. (**A**) Expression of EGFP and mCherry in HEK293T cells transfected with pPYLG + pWZG-P2A constructs and supplied with GGPP. (**B**) HPLC analysis of cells transfected with above-mentioned plasmids and supplied with GGPP. Can, canthaxanthin; Ast, astaxanthin; *E*, *trans* geometric isomer; and *Z*, *cis* geometric isomer. (**C**) Total ketocarotenoid concentration graph compared with the ratio of astaxanthin and canthaxanthin content. DW, dry weight. (**D**) Astaxanthin and canthaxanthin concentration graphs from cell extracts transfected with pPYLG + pWZG-P2A and supplied with exogenous GGPP. (**E**) The HEK293T cell pellets with a red hue reflect ketocarotenoid accumulation. (**F**) Pie charts outline total ketocarotenoids with different percentages of astaxanthin and canthaxanthin content.

**Figure 5 bioengineering-10-01073-f005:**
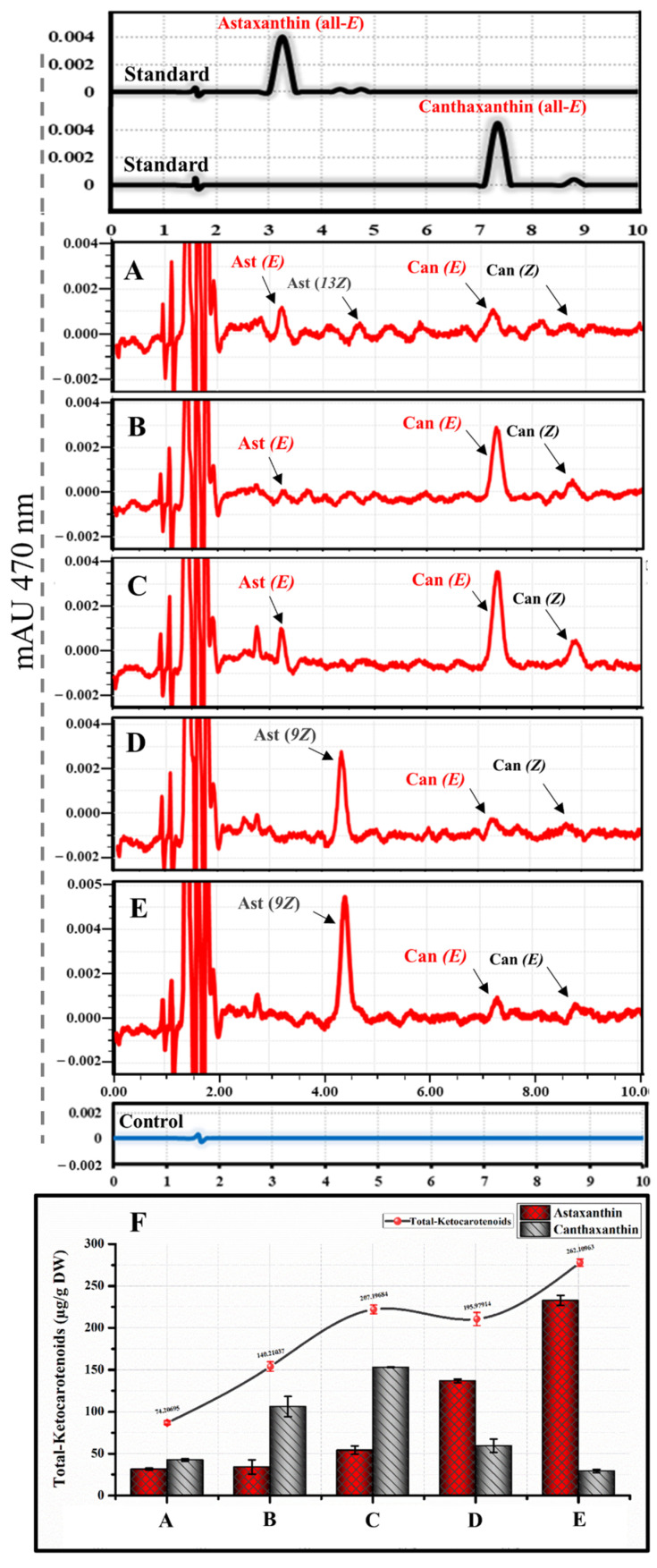
Comparative synthesis of ketocarotenoids in HEK293T cells transfected with different multiple expression plasmids. (**A**–**E**) are RP-HPLC chromatograms of ketocarotenoids for HEK293T cell extracts transfected with pWZG-A2 + β-carotene, pWZg1R-A2 + β-carotene, pPYLG + pWZG-P2A + GGPP, pAPLG + pWZG-P2A + pcrtZ-R + GGPP, and pPYLG + pWZG-P2A + pcrtZ-R + GGPP, respectively. Can, Canthaxanthin; Ast, astaxanthin; *E*, *trans* geometric isomer; and *Z*, *cis* geometric isomer. (**F**) Quantitative analysis of total ketocarotenoids (including canthaxanthin or astaxanthin). DW, dry weight. Data were presented as means ± SD (three biological replicates).

**Figure 6 bioengineering-10-01073-f006:**
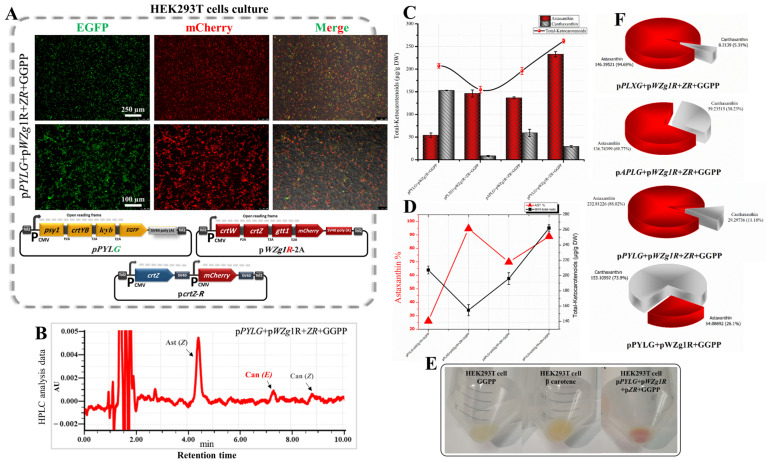
Estimate of the individual expression of *crtZ* gene for astaxanthin content enhancement in transfected HEK293T cells with β-carotene and astaxanthin biosynthetic genes. (**A**) Expression of EGFP and mCherry in HEK293T cells transfected with pPYLG + pWZG-P2A + pcrtZ-R constructs supplied with GGPP. (**B**) HPLC analysis of cells transfected with above-mentioned plasmids and supplied with GGPP. Can, canthaxanthin; Ast, astaxanthin; *E, trans* geometric isomer; and *Z, cis* geometric isomer. (**C**) Total ketocarotenoid concentration compared with the ratio of astaxanthin content. DW, dry weight. (**D**) Astaxanthin and canthaxanthin concentration graphs from cell extracts transfected with pPYLG + pWZG-P2A + pcrtZ-R plasmids and supplied with GGPP. (**E**) HEK293T cell pellets with a red color indicate the accumulation of astaxanthin. (**F**) Pie charts show total ketocarotenoids with different percentages of astaxanthin and canthaxanthin content.

**Figure 7 bioengineering-10-01073-f007:**
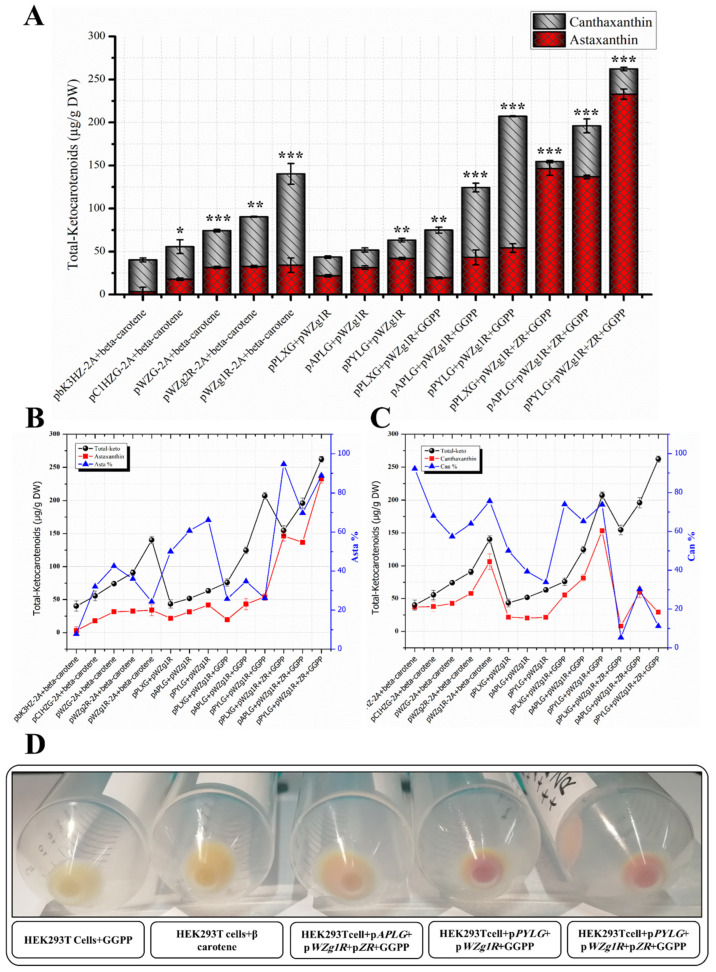
Quantitative and qualitative analytical Charts for metabolically engineered HEK293T cells using different multiple expression plasmids for astaxanthin biosynthesis. (**A**) Total ketocarotenoid concentration graph (including canthaxanthin or astaxanthin) from cell extracts transfected with different collections of plasmids. DW, dry weight. (**B**) Total ketocarotenoid concentration comparison with astaxanthin content and ratio. (**C**) Total ketocarotenoid concentration comparison with canthaxanthin content and ratio. Data were presented as means ± SD (three biological replicates), and a one-way ANOVA analysis was used to determine the significant differences between different approaches. *p*-values were calculated and represented as *p* < 0.05, *p* < 0.01, and *p* < 0.001 and indicated by *, **, and ***, respectively. (**D**) Transfected HEK293T cell pellets with red color indicate the accumulation of ketocarotenoids compared with control samples.

**Table 1 bioengineering-10-01073-t001:** Compositions and contents of total ketocarotenoids, including canthaxanthin and astaxanthin, in transfected HEK293T cells. Data (µg/g DW, gram of dry weight) were presented as means ± SD of three biological replicates.

Samples	Ast ± SD	Ast %	Can ± SD	Can %	Total (Ast + Can)
pbK3HZ-2A + β-carotene	3.1 ± 5.4	7.8	37.1 ± 2.4	92.2	40.2 ± 7.6
pC1HZG-2A + β-carotene	17.8 ± 1.5 **	32.0	37.8 ± 8.0 *	67.9	55.6 ± 7.3
pWZG-2A + β-carotene	31.6 ± 1.3 **	42.6	42.6 ± 1.5 ***	57.4	74.2 ± 2.2
pWZg2R-2A + β-carotene	32.5 ± 1.3	35.9	57.9 ± 0.6	64.0	90.4 ± 1.8 **
pWZg1R-2A + *β-carotene*	34.1 ± 8.5	24.3	106.1 ± 12.0 ***	75.7	140.2 ± 5.5 ***
pPLXG + pWZg1R	21.7 ± 5.1	49.9	21.8 ± 1.3	50.0	43.5 ± 5.9
pAPLG + pWZg1R	31.4 ± 2.0 *	60.7	20.3 ± 2.5	39.3	51.7 ± 1.7
pPYLG + pWZg1R	41.9 ± 1.3 ***	66.2	21.4 ± 2.1	33.8	63.2 ± 3.1 **
pPLXG + pWZg1R + GGPP	19.5 ± 1.1 ***	25.7	55.4 ± 3.5	73.9	74.9 ± 4.2 **
pAPLG + pWZg1R + GGPP	43.2 ± 8.7 ***	34.7	81.2 ± 5.1 *	65.3	124.4 ± 5 ***
pPYLG+pWZg1R+GGPP	54.1 ± 5.0 *	26.1	153.1 ± 0.5 ***	73.9	207.2 ± 5.1 ***
pPLXG + pWZg1R + pZR + GGPP	146.4 ± 7.8 ***	94.7	8.2 ± 1.2 ***	5.3	154.6 ± 7.3 ***
pAPLG + pWZg1R + pZR + GGPP	136.7 ± 2.1 ***	69.8	59.2 ± 8.0 **	30.2	196.0 ± 7.7 ***
pPYLG + pWZg1R + pZR + GGPP	232.8 ± 6.0 ***	88.8	29.3 ± 2.0 **	11.2	262.1 ± 4.3 ***

*p* < 0.05, *p* < 0.01, and *p* < 0.001 are tagged as *, **, and ***, respectively. The astaxanthin and canthaxanthin are the sum of E (trans) and Z (cis) isomers. Can, canthaxanthin; Ast, astaxanthin; and SD, standard deviation. Calculated *p*-values are presented in Appendix A.

## Data Availability

The data presented in this study are available in the GenBank data libraries (https://www.ncbi.nlm.nih.gov/genbank/, accessed on 15 April 2023) under the following accession numbers: MT360309 (crtW), MT360311 (crtZ), OR402794 (bkt3), OR402795 (H-crtZ), OR402796 (cbkI), OR402797 (DGTT1), OR402798 (DGTT2), OR402799 (psy1), OR402800 (lcyb), OR402801 (Pa-crtI), OR402802 (Xd-crtI), and OR402803 (crtYB).

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
