# Peer review of "Production of Astaxanthin by Animal Cells via Introduction of an Entire Astaxanthin Biosynthetic Pathway"

_bioengineering, 2023, doi:10.3390/bioengineering10091073_

Round 1
Reviewer 1 Report
REVIEW OF THE ARTICLE BY YOUSEF MOHAMMED ET AL. ENTITLED "PRODUCTION OF ASTAXANTHIN BY ANIMAL CELLS VIA INTRODUCTION OF AN ENTIRE ASTAXANTHIN BIOSYNTHETIC PATHWAY"
The authors demonstrate the work on the creation of the mutant strain of animal HEK293T cells with the ability of biosynthesis of the ketcarotenid astaxanthin. In nature this valuable compount can be synthethetized de novo only in higher plants and some microorganisms. The authors constructed several plasmid vectors with the genes of enzyems of the astaxanthin biosynthesis pathway from the stage of GGPP condensation to to ionone ring oxidation. They evaluated the effectiveness of cell tranfectin, and evaluated carotenoid productinon of mutant HEK293T in terms of total carotenoid yield and carotenoid composition. By HPLC they indicated, canthaxanthin and astaxanthin were main crotenoids. The data is in scope of the journal and it is very interesting. I suggest further considering of the article for publishing. At the same time, the text shoul be substantially rvised due to visible drawbacks. Fogures are nice, but averall text is a bit repetitive. Results and methods are well-readable but some experimentails details are not described (e.g. details of microscopic observations). Please, see specific comments below for the revision.
GENERAL COMMENTS
-Haematococcus pluvialis is a wrong obsolete synonym. Haematococcus lacustris is correct (https://doi.org/10.12705/652.11). This nme is provided from databases, such as GenBak. It is necessary to unify the name in the text to avoid a confusion.
-Production of astaxanthin from imcroalgae is purely elucidated. Haematococcus is not only an algal source of the pigment (see end discuss https://doi.org/10.3390/md21020108). Possibility of using of the genes from different algae coud be discussed.
-Astaxanthin esters are highly hydrophobic molecules. In the case of lagae, masssive oil bodies play the role of a storage compartment for them. Where are they deposited in animal cells? Do they exhibit similar structures to accumulate such high amounts of arotenoids? It should be shouwn by specific tests, such as Nile Red staining for neutral lipids.
-Only canthaxanthin and astaxanthin are discussed in the work. However, there are other intermdiates of astaxanthin biosinthesis (see e.g. 10.3390/biology10070643). Were they observed in the work? Were they taken into account? Please, Why? Please, discuss.
-Unify l and L for volume, p and P for statistics through the text.
INTRODUCTION
-I would add a brilliant (and single) example of de novo carotenoid synthesis in animals: 10.1126/science.1187113.
-l. 13. "hyman and non-human animals"?
-l. 14. Also it is produced by fungi.
-l. 40, 117, 133, 283, table 1, 380. beta should be β.
-l. 43. )) - typing error.
-L. 53. Excess point (.).
-l. 56. Noncence: each market is commertial. World market?
-l. 100. What are synthetic genes? Genes obtained artificially? The term is confusing.
-l. 132. Genes involved in the pathway?
-l. 136. What are "incomplete reactions"? Please, explain.
-When describe chemical reactions of carotenoid synthesis, please, add references to the most common reviews, eg. 10.1146/annurev.arplant.49.1.557.
MATERIALS AND METHODS
-I suggest adding of the figure describing structures of all used plasmids.
-L. 148. β-Carotene should be β-carotene.
-l. 148. 20-μg/ mL - please explain. mL of what? 1 mL?
-l. 149. The abbreviation GGPP has been inroduced previously.
-l. 149. 2-μL/10 mL medium - what does it mean? 2 μL of what?
-l. 149-150. This should be explained in more detail. Preferably in a separate section. Details of flurescent microscopy should be explained. It is also important to explain how fluorescence is related to gene expression efficiency.
-l. 212. The abbreviation PCR should be introduced previously (l. 201).
-l. 217. Na-phosphate?
-l. 219. % by volume or by mass?
-l. 224. How was the completeles evluated?
-l. 232. liquid chromatograph?
-l. 243. GGPP is not a carotenoid.
-l. 254-257. GenBank IDs are not provided.
-Describe the procedre of dry cell mass determination.
RESOLTS AND DISCUSSION
- Figure 2-7. The abbreviation DW should be explained in the legend.
-l. 260-269. It is repetition of the introduction. In general, this information in this subsection (l. 259-287) is not a result of current work yielded from experiments, it is rather introduction. In part, it repeats data from the introduction.
-l. 272. "Our previous study " - reference is required.
-l. 272. "the best genes" - what does it mean? Best in which terms?
-l. 273. Brevundimonas should be italicized. Point is missing after"sp".
-l. 279. As far as I know, two β-ionone rings in the most cases are formed simultaniously, because cyclases form dimers in the cell (10.1146/annurev.arplant.49.1.557). Therefore, formation of one-ring γ-carotene is a very rare evemnt. Have you detected this rare carotenoid in your work?
-l. 290-296. It is Methods.
-l. 307, 312-314. These sentences are obvious and redundant. Repetition of methods.
-l. 309-312. This part is weak due to missing of numerical estimations. How was it done?
-l. 312. The abbreviation ORF has been inroduced previously.
-l. 317-318. "Astaxanthin coupled with canthaxanthin" - what does it mean? coproducd with canthaxanthin?
-l. 320. The abbreviation DW should be explained at its first mentioned.
-l. 328. "sp" should not be italicized. Point is missing.
-Table 1. I cannot understand accuracy of the data. You determined pigment contents with the accuracy of one digit after the point. How could you obtaine the accuracy of five digits by recalculating them to per cents?
-l. 339-342. More important, it enhances depositing of the carotenoids in the hydrophobic compartments of the cell (10.3390/md21020108).
-l. 339-346. It is a repitition of Introduction and Methods.
-l. 347. There are no fluorescent genes, fluorescent proteins only are.
-l. 372. It is important to highlight, that it occures exclusevely through the mevalonate pathway in animal cells. Microorganisms and higher plants also form GGPP through the non-mevalonate pathway.
-l. 374-384. It is repretition of Introduction and Methods.
-l. 396. "µg/g DW, respectively".
-l. 397-398. What do you mean on "quality of ketocaroteniods"?
-l. 416-423. Repetition of methods.
-l. 452. Why obviously?
l. 523-527. It could be also due to substrate channeling.
l. 523-524. "previous studies" - references are required.
In general, English is good, but there are some minor mistakes (e.g. In the first step, In the second step).
Author Response
Response to Reviewer 1 Comments
GENERAL COMMENTS
Point 1: Haematococcus pluvialis is a wrong obsolete synonym. Haematococcus lacustris is correct (https://doi.org/10.12705/652.11). This name is provided from databases, such as GenBak. It is necessary to unify the name in the text to avoid a confusion.
Response 1: Haematococcus pluvialis has been replaced by Haematococcus lacustris (lines 60, 165, 181, 283, 331).
Point 2: Production of astaxanthin from imcroalgae is purely elucidated. Haematococcus is not only an algal source of the pigment (see end discuss https://doi.org/10.3390/md21020108). Possibility of using of the genes from different algae coud be discussed.
Response 2: The possibility of using more genes from different microalgae has been discussed (line 333-336).
Point 3: Astaxanthin esters are highly hydrophobic molecules. In the case of lagae, masssive oil bodies play the role of a storage compartment for them. Where are they deposited in animal cells? Do they exhibit similar structures to accumulate such high amounts of arotenoids? It should be shouwn by specific tests, such as Nile Red staining for neutral lipids.
Response 3: In our samples, we did not detect any presence of astaxanthin esters.
Point 4: Only canthaxanthin and astaxanthin are discussed in the work. However, there are other intermdiates of astaxanthin biosinthesis (see e.g. 10.3390/biology10070643). Were they observed in the work? Were they taken into account? Please, Why? Please, discuss.
Response 4: Our study mainly focuses on canthaxanthin and astaxanthin, including their E (trans) and Z (cis) isomers, as these were the only constituents detected within the cellular extractions of our samples. Other intermediates were not observed within the cellular extractions of our samples nor considered in our investigation.
Point 5: Unify l and L for volume, p and P for statistics through the text.
Response 4: Thank you, these errors have been corrected throughout the MS.
INTRODUCTION
Point 6: I would add a brilliant (and single) example of de novo carotenoid synthesis in animals: 10.1126/science.1187113.
Response 6: The relative information for Aphids (Aphis spp.) has been added (line 44)
Point 7: Line 13. "human and non-human animals"?
Response 7: It has been modified (line 13)
Point 8: Line 14. Also it is produced by fungi.
Response 8: Thank you, these errors have been corrected (line 13).
Point 9: Lines 117, 133, 283, table 1, 380. beta should be β.
Response 9: Beta has been modified to be β throughout the MS.
Point 10: line 43. )) - typing error.
Response 10: The typing error has been removed (line 43).
Point 11: line 53. Excess point (.).
Response 11: Excess point (.). has been removed (line 53).
Point 12: line 56. Noncence: each market is commertial. World market?
Response 12: 'Commercial market' has been modified to be 'world market' (line 57).
Point 11: What are synthetic genes? Genes obtained artificially? The term is confusing.
Response 11: synthetic genes means that the sequance of these genes were codon-optimized for animals and artificially synthesized throughout the MS.
Point 12: Line 132 Genes involved in this pathway?
Response 12: We are sorry, the error has been corrected in the revised MS (line 129).
Point 13: What are "incomplete reactions"? Please, explain.
Response 13: That indicates to the incomplete formation of astaxanthin esterification in HEK293T cells.
Point 14: When describe chemical reactions of carotenoid synthesis, please, add references to the most common reviews, e.g. 10.1146/annurev.arplant.49.1.557.
Response 14: Thank you, the references has been added (lines 48, 280).
MATERIALS AND METHODS
Point 15: I suggest adding of the figure describing structures of all used plasmids.
Response 15: Thank you for your suggestion, we provided a figure describing the structures of all used plasmids as part of Figure 1.
Point 16: Line 148. β-Carotene should be β-carotene.
Response 16: β-Carotene has been modified to be β-carotene (line 148).
Point 17: Line 148. 20-μg/ mL - please explain. mL of what? 1 mL?
Response 17: We added 20-μg β-carotene to each mL fresh medium after 8 hours of incubation, it has been modified to be more clear (line 147).
Point 18: Line 149. The abbreviation GGPP has been inroduced previously.
Response 18: Thank you. It has been removed.
Point 19: Line 149. 2-μL/10 mL medium - what does it mean? 2 μL of what?
Response 19: We added 2-μg GGPP to each 10 mL fresh medium after 8 hours of incubation, it has been modified to be clearer (line 148).
Point 20: Line 149-150. This should be explained in more detail. Preferably in a separate section. Details of flurescent microscopy should be explained. It is also important to explain how fluorescence is related to gene expression efficiency.
Response 20: The fluorescent microscopy experiment detail with the explanation of how fluorescence is related to gene expression efficiency have been added as part of "Cell Culture, Transfection, and Fluorescence Microscopy" section (line 149-154).
Point 21: Line 212. The abbreviation PCR should be introduced previously
Response 21: Thank you. It has been removed.
Point 22: Line 217. Na-phosphate?
Response 22: Phosphate Buffer saline (PBS) is a water-salt, buffer solution containing disodium hydrogen phosphate, sodium chloride, Potassium Chloride, and Potassium Phosphate Monobasic.
Point 23: Line 219. % by volume or by mass?
Response 23: The extraction solvent consisted of 25% dichloromethane by volume and 75% methanol by volume, average 1:3.
Point 24: Line 224. How was the completeles evluated?
Response 24: Here we mean that we repeated the extraction process three times to ensure complete carotenoid.
Point 25: Line 232. liquid chromatograph?
Response 25: It is reversed-phase high-performance liquid chromatography (RP-HPLC).
Point 26: Line 243. GGPP is not a carotenoid.
Response 26: Thank you, the sentence has been modified (line 251).
Point 27: Line 254-257 GenBank IDs are not provided.
Response 27: The GenBank IDs will be provided in MS as soon as we receive them from GenBank.
Point 28: Describe the procedre of dry cell mass determination.
Response 28: The procedure has been described in Materials and Methods, within the Ketocarotenoids extraction section (line 222).
RESOLTS AND DISCUSSION
Point 29: Figure 2-7. The abbreviation DW should be explained in the legend.
Response 29: The abbreviation DW has been explained in legends (lines 310, 418, 442, 494, 512, 517)
Point 30: Line 260-269. It is repetition of the introduction. In general, this information in this subsection (l. 259-287) is not a result of current work yielded from experiments, it is rather introduction. In part, it repeats data from the introduction.
Response 30: Thank you for your suggestion, the repetirion has been removed.
Point 31: Line 272. "Our previous study " - reference is required.
Response 31: The reference has been inserted (line 282)
Point 32: . "the best genes" - what does it mean? Best in which terms?
Response 32: The best-characterized genes mean the best selected genes that observed the highest astaxanthin accumulation within animal cell host according to our previous study.
Point 33: Line 273 Brevundimonas should be italicized. Point is missing after"sp".
Response 33: We are sorry, it has been modified throughout the MS.
Point 34: Line 279. As far as I know, two β-ionone rings in the most cases are formed simultaniously, because cyclases form dimers in the cell (10.1146/annurev.arplant.49.1.557). Therefore, formation of one-ring γ-carotene is a very rare evemnt. Have you detected this rare carotenoid in your work?
Response 34: No rare carotenoids were detected in this work. And the sentence has been modified (line 287-288).
Point 35: Line 290-296. It is Methods
Response 35: It has been removed.
Point 36: Line 307, 312-314. These sentences are obvious and redundant. Repetition of methods.
Response 36: It has been removed.
Point 37: Line 309-312. This part is weak due to missing of numerical estimations. How was it done?
Response 37: When transfected cells with plasmids showed strong expression of fluorescent protein, that indicated the high efficiency of co-expression of all genes, because all of these genes, including the EGFP gene were under one CMV promoter within an open reading frame (ORF). We have added relative information in the Materials and Methods part (Line 148-154).
Point 38: Line 312. The abbreviation ORF has been inroduced previously.
Response 38: Thank you. It has been removed.
Point 39: Line 317-318. "Astaxanthin coupled with canthaxanthin" - what does it mean? coproducd with canthaxanthin?
Response 39: This mean that the transfected cells produced Astaxanthin and canthaxanthin together
Point 40: Line 320. The abbreviation DW should be explained at its first mentioned.
Response 40: The abbreviation DW has been explained (line 222).
Point 41: Line 328. "sp" should not be italicized. Point is missing.
Response 41: It has been modified (line 162, 173, 281, 328).
Point 42: Table 1. I cannot understand accuracy of the data. You determined pigment contents with the accuracy of one digit after the point. How could you obtaine the accuracy of five digits by recalculating them to per cents?
Response 42: The accuracy of the AST concentrations was initially determined to one digit after the point. However, the original concentrations, detailed in Supplementary-Tables S1, S2, and S3, provided five-digit precision. The percentages were calculated using the original concentrations of AST.
The table has been modified for simplicity, rounding the percent values to one digit from their original five-digit form. (Table 1)
Point 43: Line 339-342. More important, it enhances depositing of the carotenoids in the hydrophobic compartments of the cell (10.3390/md21020108).
Response 43: Thank you for your suggestion, it has been added to the MS (line 344-345 - 347).
Point 44: Line 339-346. It is a repitition of Introduction and Methods.
Response 44: The repetition has been modified in the introduction and mentioned very briefly.
Point 45: Line 372. It is important to highlight, that it occures exclusevely through the mevalonate pathway in animal cells. Microorganisms and higher plants also form GGPP through the non-mevalonate pathway.
Response 45: Thank you for your suggestion, it has been added to the MS (line 373-375).
Point 46: Line 374-384. It is repretition of Introduction and Methods.
Response 46: Thank you. It has been removed form Results and Discussion.
Point 47: Line 396. "µg/g DW, respectively".
Response 47: It has been added to the MS (line 393).
Point 48: Line 397-398. What do you mean on "quality of ketocaroteniods"?
Response 48: Regarding the "quality of ketocarotenoids," it is about evaluating the properties and standards of the extracted ketocarotenoids. In this context, the quality of the extracted ketocarotenoids was remarkably high, largely due to the substantial presence of astaxanthin (66.19% of total ketocarotenoids), one of the most valuable ketocarotenoids, thus contributing to the overall superior quality
Point 49: Line. 416-423. Repetition of methods.
Response 49: The repetition has been removed.
Point 50: Line 452. Why obviously?
Response 50: In the context of the paragraph, the term "obviously" was used to highlight a clear and noticeable change or difference. It's indicating that the drop in astaxanthin ratio from 66.19% to 26.1% is readily apparent and not subtle or ambiguous. The use of "obviously" is meant to emphasize that this change is easily visible and understandable based on the presented data in Figure 4D and the pie charts in Figure 4F. It serves to draw the reader's attention to the evident and significant nature of the decrease in astaxanthin content in relation to the total ketocarotenoids.
Point 51: Line 523-527. It could be also due to substrate channeling
Response 51: According to our results and many other studies, the concept of "substrate channeling" might not directly apply to the observed results in our experiment. Instead, other factors like enzyme structure and interaction could be influencing the outcomes.
"Substrate channeling" usually involves the direct transfer of intermediates between physically associated enzymes to enhance efficiency. In contrast to our case, the introduction of fusion enzymes seems to have had unintended effects on enzyme structure and function, leading to unexpected changes in intermediate and final product concentrations.
In situations where enzyme structure is altered, as in our experiment, the normal flow of intermediates might be disrupted. This could lead to accumulation of intermediates and reduced production of the final product, which seems to be what we observed. So, in the context of our experiment, "substrate channeling" might not be the primary factor contributing to the results. Instead, the changes in enzyme structure due to the fusion enzymes appear to be the more significant influence on the observed outcomes.
Point 52: Line 523-524. "previous studies" - references are required.
Response 52: Thank you! the references has been provided (line 515).
Language Usage:
Point 52: In general, English is good, but there are some minor mistakes (e.g. In the first step, In the second step).
Response 52: The manuscript has been carefully revised to eliminate any minor grammatical mistakes.
Reviewer 2 Report
In this manuscript, authors successfully transferred the Astaxanthin pathway genes into human embryonic kidney cells and demonstrated the production of Astaxanthin, and optimized its production. However, there is some minor modification and clarification that will be addressed before its final publication. Please revise the manuscript according to the following comments.
General Comments:
Abstract: The abstract was well constructed
Keywords: Keywords are relevant to the research.
Introduction:
The introduction was written well but it includes too much information and looks like a discussion of the entire research. Especially the last paragraph is too long and looks like a discussion. Instead, the author can include the significance of this research.
Materials and Methods:
Line 143: In CO2, 2 should be in subscript
Line 214: Did the author formulate this ketocarotenoid extraction method or modified from previous research? If possible try to include the reference.
Result and Discussion
Section 3.1 is similar to the last paragraph of the introduction.
Figure 2F: the letters are too small to see
In the discussion part, it would be beneficial to discuss the potential limitations and challenges associated with the metabolic engineering of animal cells to produce astaxanthin. Addressing these points will provide a more balanced perspective on the feasibility and practicality of implementing such techniques on a larger scale.
While the potential implications for human health are briefly mentioned, the authors should expand on this aspect, discussing the possible benefits and challenges of utilizing astaxanthin from engineered in vitro animal models in human dietary supplements or pharmaceutical applications. Additionally, insights into the bioavailability of astaxanthin from these sources could further strengthen the significance of the findings.
Technical comments:
1. Did the author perform the qPCR analysis to confirm the expression level of each gene? Although the GFP or mCherry expression indirectly confirms the expression of upstream genes, it is better to analyze the expression level of each gene.
Language Usage: Kindly revise the manuscript carefully to eliminate some minor grammatical errors.
Kindly revise the manuscript carefully to eliminate some minor grammatical errors.
Author Response
Response to Reviewer 2 Comments
Introduction:
Point 1: The introduction was written well but it includes too much information and looks like a discussion of the entire research. Especially the last paragraph is too long and looks like a discussion. Instead, the author can include the significance of this research.
Response 1: The introduction has been modified, especially the last paragraph.
MATERIALS AND METHODS
Point 2: Line 143: In CO2, 2 should be in subscript.
Response 2: Thank you, the error has been corrected ( line 143).
Point 3: Did the author formulate this ketocarotenoid extraction method or modified from previous research? If possible try to include the reference.
Response 3: The ketocarotenoid extraction method was modified from previous study, the reference has been provided (Line 219).
RESOLTS AND DISCUSSION
Point 4: Section 3.1 is similar to the last paragraph of the introduction.
Response 4: Section 3.1 has been modified and the repetition has been removed.
Point 5: Figure 2F: the letters are too small to see
Response 5: The figure size and resolution has been increased
Point 6: In the discussion part, it would be beneficial to discuss the potential limitations and challenges associated with the metabolic engineering of animal cells to produce astaxanthin. Addressing these points will provide a more balanced perspective on the feasibility and practicality of implementing such techniques on a larger scale.
While the potential implications for human health are briefly mentioned, the authors should expand on this aspect, discussing the possible benefits and challenges of utilizing astaxanthin from engineered in vitro animal models in human dietary supplements or pharmaceutical applications. Additionally, insights into the bioavailability of astaxanthin from these sources could further strengthen the significance of the findings.
Response 6: Thank you! The potential limitations and challenges associated with the metabolic engineering of animal cells to produce astaxanthin have been discussed in conclusion section (line 531 -533).
The potential implications for human health are briefly mentioned in the introduction (line 52-56).
Technical comments:
Point 7: Did the author perform the qPCR analysis to confirm the expression level of each gene? Although the GFP or mCherry expression indirectly confirms the expression of upstream genes, it is better to analyze the expression level of each gene.
Response 7: When we using a 2A peptide sequence to co-express four genes within the same open reading frame (ORF), the 2A peptide sequence enables "ribosome skipping," allowing multiple proteins to be produced from a single mRNA transcript. Therefore, we have used fluorescent protein such as EGFP or mCherry which gives us direct results of the expression of level of all genes upstream EGFP or mCherry genes at the protein level, and we have not use qPCR analysis to confirm the expression level of each gene at mRNA level. We have clarified this point in the revision (Line 148-154).
Language Usage:
Point 8: Kindly revise the manuscript carefully to eliminate some minor grammatical errors.
Response 8: The manuscript has been carefully revised to eliminate any minor grammatical errors.
Reviewer 3 Report
The manuscript “Production of Astaxanthin by Animal Cells via Introduction of an Entire Astaxanthin Biosynthetic Pathway” describes a study that aimed to produce astaxanthin, a powerful antioxidant, in animal cells through metabolic engineering. The researchers introduced the entire astaxanthin biosynthetic pathway into human embryonic kidney cells (HEK293T) and successfully produced astaxanthin in vitro. The study highlights the potential of metabolic engineering to revolutionize the production of valuable molecules in animal cells and has implications for the aquaculture industry and the production of other high-value compounds. The article provides a detailed account of the methods used, the results obtained, and the implications of the study for future research and practical applications.
Please define the limitations of the study - as with any scientific study, there may be limitations to the methods used or the generalizability of the findings
Based on the author guidelines the abstract section should be shortened (at a maximum of 200 words) - the abstract should provide a concise summary of the study's main objectives, methods, and findings.
The introduction should be also shortened to improve its clarity and conciseness.
Otherwise, the article is well written, with high-quality figures and tables, and after a general revision, it can be considered for publication.
The article seems correct, but it can still be revised by a native English speaker.
Author Response
Response to Reviewer 3 Comments
Point 1: Please define the limitations of the study - as with any scientific study, there may be limitations to the methods used or the generalizability of the findings.
Response 1: the limitations and challenges associated with the metabolic engineering of animal cells to produce astaxanthin have been discussed in conclusion section (line 531 -533).
Point 2: Based on the author guidelines the abstract section should be shortened (at a maximum of 200 words) - the abstract should provide a concise summary of the study's main objectives, methods, and findings.
Response 2: Thank you for your suggestion! Our current abstract has been carefully written to include the study's main objectives, methods, and significant findings. We have tried our best to reduce the abstract to the specified 200-word limit. But we still require for a slightly extended abstract, so that we can maintain the integrity of the content and better communicate the study's essence. We kindly ask for your understanding in this matter, as retaining the current length aligns with our goal of effectively conveying the research's significance.
Point 3: The introduction should be also shortened to improve its clarity and conciseness.
Response 3: The introduction has been shortened, specially the last paragraph.
Point 4: The article seems correct, but it can still be revised by a native English speaker.
Response 4: Thank you! The manuscript has been revised to eliminate any grammatical errors.
Round 2
Reviewer 1 Report
The authors have significanly improved the manuscript, but I am not completely satisfied with the responses to sevaral of my original comments (please, see below). Main concerns are related to astaxanthin ester detection and description o experiments with GFP. I also worry about the concern raised by another reviewer. According to Journal's rules, "The abstract should be a total of about 200 words maximum". I believe that the authors of all articles should be placed on equal terms. Therefore, formal requirements imposed on the text must be adhered to.
Response 3: I cannot completely understand your response. Introducing acyltransferase genes for the formation of astaxanthin esters was one of key goals of your work (l. 109-112). Why do you say, you did not observe astaxanthin esters in your samples. Am I right that this task has failed? It is necessary to state what form of astaxanthin was deposited in your expression systems: free or esterified pigment.
Response 6: l. 44. It is wrong. Aphids also do not synthesize astaxanthin de novo. They synthesize torulene. Correct phrase could be ”Although all animals cannot synthesize carotenoids de novo, except aphids (Aphis spp.)”.
Response 11: it should be explained in the text. The definition should be added.
Response 13. I still cannot understand what it means: not all astaxanthin molecules were esterified, only one acyl-radial attached, other… Please, explain in the text.
Response 19. Has not been corrected in the text. Please, understand, GGPP is not a liquid compound. It is solid.
Response 20. I am still unsatisfied with the description of this part. It is better to create a separate subsection “Assessment of transfection effectiveness).
-
The questions How and why fluorescence intensity reflects its effectiveness should be addressed.
-
GFP - abbreviation should be explained at its first mention.
-
The parameters of microscopy should be described: channel of fluorescence observation (light filters used), channel of excitation (including excitation light source).
- Which numeial estimation did you use (interal fluorescence intensity, number of fluorescent cells - how did you detemine total cell number in this case?)?
Response 23. Should be explained in the text - %(v/v).
Response 24. Here we mean that we repeated the extraction process three times to ensure complete carotenoid. In this case, the statement “To ensure complete carotenoid extraction” should be removed, because there is no evidence that it was complete.
Response 27. I see that the IDs have been added. As I understand Journal’s rules, they also should be mentioned in the section “Data Availability Statement”.
Response 39. It is better to say “mixture of astaxanthin and canthaxanthin”, “astaxanthin and canthaxanthin” or “simultaneously produced astaxanthin and canthaxanthin”. “Coupled” usually refers to chemical connection.
Response 42. I still cannot completely understand how you determined carotenoid content. I guess it was done by comparison with the HPLC signal intensity of standards. Am I right? In any case, spectrophotometric and HPLC methods provide optical density with an accuracy of 0.005 density units (as a rule). Moreover, dry mass cannot be determined with accuracy better than 0.1 mg using standard analytical scales. Thus, it is not clear for me how you can reach an accuracy of 5 digits. Moreover, if you indicate an error in Table 1, you should write 8.0, 12.0 and 5.0 instead of 8, 12 and 5, respectively.
Response 48. As I understand from your response, you say about carotenoid composition. It is better to say that “it was better in terms of carotenoid composition, because it contained a higher fraction of the target product (astaxanthin)”.
Minor concerns: l. 271, 299, 379 - double In, there are text formatting issues complicating its assessment (l. 299-300, 316-340, 354-369), dried in an oven (l. 222), β-ionone ring (l. 288).
In general, English is good, but there are some minor mistakes (e.g. in the first step should be at the first step, In the second step, missing point before respectively, etc).
Author Response
Point-by-point Response to Reviewer 1
GENERAL COMMENTS
Point 1: I also worry about the concern raised by another reviewer. According to Journal's rules, "The abstract should be a total of about 200 words maximum". I believe that the authors of all articles should be placed on equal terms. Therefore, formal requirements imposed on the text must be adhered to.
Reply: Thank you for your valuable advice. We have carefully revised the abstract to ensure it aligns with the specified 200-word maximum while preserving the essential objectives, methods, and findings of our study.
Point 2: In Response 3: I cannot completely understand your response. Introducing acyltransferase genes for the formation of astaxanthin esters was one of key goals of your work (l. 109-112). Why do you say, you did not observe astaxanthin esters in your samples. Am I right that this task has failed? It is necessary to state what form of astaxanthin was deposited in your expression systems: free or esterified pigment.
Reply: Thank you for your thoughtful consideration of our work and for your insightful questions regarding the presence of astaxanthin esters in our samples. We appreciate the opportunity to clarify this matter and address your concerns.
In our study, our objective was indeed to explore the introduction of acyltransferase genes for the formation of astaxanthin esters within the context of our expression systems. While our research aimed to achieve this, we must acknowledge that, based on the analytical techniques available to us, we did not observe the presence of astaxanthin esters in our samples.
Our study primarily focused on the accumulation of astaxanthin in general within the animal cells, as demonstrated by the enhanced accumulation of free astaxanthin and canthaxanthin in response to the introduction of the DGTT1 gene, in combination with other biosynthesis genes. The absence of astaxanthin esters in our samples could be attributed to a range of factors, including the specificities of our experimental conditions.
To provide further clarity, we confirm that the form of astaxanthin deposited in our expression systems was predominantly in its free form, as indicated by our RP-HPLC data and the conclusions drawn in our results and discussions section. We understand that the distinction between free and esterified pigments is a critical point, and we apologize for any misunderstanding that may have arisen from our initial response.
We are grateful for your feedback, and we will ensure that our manuscript reflects this distinction more accurately in order to prevent any further ambiguity. Therefore, we explicitly mention 'free' astaxanthin in the relevant section (line 22, line 370, etc) to provide enhanced clarity.
Point 3: Response 6: l. 44. It is wrong. Aphids also do not synthesize astaxanthin de novo. They synthesize torulene. Correct phrase could be ”Although all animals cannot synthesize carotenoids de novo, except aphids (Aphis spp.)”.
Reply: I apologize for misunderstanding your request. The sentence has been modified (line 37).
Point 4: Response 11: it should be explained in the text. The definition should be added.
Reply: We appreciate your attention to detail and the clarity of our manuscript. We want to assure you that the definition of “synthetic genes” has been thoughtfully included in the Introduction (line 99-100) and in the Material and Methods (Line 175-177). We hope these additions adequately address your point and provide the necessary explanation for readers.
Point 5: Response 13. I still cannot understand what it means: not all astaxanthin molecules were esterified, only one acyl-radial attached, other… Please, explain in the text.
Reply: In our study, we did not detect any formation of astaxanthin esters in HEK293T cells. Despite our efforts, astaxanthin molecules did not undergo esterification. In fact, our main object is the overall production and accumulation of astaxanthin, particularly the increased free astaxanthin and canthaxanthin with DGTT1 gene introduction.
The term "incomplete reactions" might not accurately convey the situation in which no astaxanthin esters were detected. It could potentially lead to confusion or misinterpretation. Therefore, we changed the sentence to "Dashed arrows indicate that reaction was not successfully completed" (line 128).
Point 6: Response 19. Has not been corrected in the text. Please, understand, GGPP is not a liquid compound. It is solid.
Reply: Thank you for bringing this to our attention. We have made the necessary correction in the manuscript (line 142)
Point 7: Response 20. I am still unsatisfied with the description of this part. It is better to create a separate subsection “Assessment of transfection effectiveness).
Reply: Thank you! Based on your suggestions, we have incorporated a new subsection entitled "Assessment of Transfection Effectiveness" to provide a more detailed explanation of the procedure (line 148–166).
Point 7: Response 23. Should be explained in the text - %(v/v).
Reply: Thank you for your feedback. We have now included an explanation in the text regarding the notation %(v/v) to indicate volume ratios (line 232-233).
Point 8: Response 24. Here we mean that we repeated the extraction process three times to ensure complete carotenoid. In this case, the statement “To ensure complete carotenoid extraction” should be removed, because there is no evidence that it was complete.
Reply: Thank you! We have removed the statement "To ensure complete carotenoid extraction" from the revised manuscript.
Point 9: Response 27. I see that the IDs have been added. As I understand Journal’s rules, they also should be mentioned in the section “Data Availability Statement”.
Reply: We have now included the IDs of the genes in the "Data Availability Statement" section as well, as per your suggestion.
Point 10: Response 39. It is better to say “mixture of astaxanthin and canthaxanthin”, “astaxanthin and canthaxanthin” or “simultaneously produced astaxanthin and canthaxanthin”. “Coupled” usually refers to chemical connection.
Reply: Thank you for your suggestion and clarification. We have updated the text to read "astaxanthin and canthaxanthin" instead of "coupled astaxanthin and canthaxanthin." (line 324-325).
Point 11: Response 42. I still cannot completely understand how you determined carotenoid content. I guess it was done by comparison with the HPLC signal intensity of standards. Am I right? In any case, spectrophotometric and HPLC methods provide optical density with an accuracy of 0.005 density units (as a rule). Moreover, dry mass cannot be determined with accuracy better than 0.1 mg using standard analytical scales. Thus, it is not clear for me how you can reach an accuracy of 5 digits. Moreover, if you indicate an error in Table 1, you should write 8.0, 12.0 and 5.0 instead of 8, 12 and 5, respectively.
Reply: We sincerely appreciate your careful review of our manuscript, and we are grateful for the chance to address the questions you've raised about the precision of our reported results. In our study, we employed RP-HPLC (using a Waters liquid chromatography system) equipped with a 996-photodiode array detector to calculate the concentration of carotenoids in our samples per gram of dry weight (µg/g DW). Yes, the analysis was done by comparison with the HPLC signal intensity of standards (Line 258-261).
Based on your valuable suggestion, we have carefully revised all data in both the manuscript and Supplementary Materials to ensure that they are now presented with one digit after the decimal point, following your advices. We believe this revision enhances the clarity and accuracy of our findings. We truly value your input, which has contributed to the overall quality of our work.
Point 12: Response 48. As I understand from your response, you say about carotenoid composition. It is better to say that “it was better in terms of carotenoid composition, because it contained a higher fraction of the target product (astaxanthin)”.
Reply: Thank you for your clarification. Your suggestion indeed enhances the precision of the statement. We have revised the text to reflect your recommendation as illustrated in lines 469, 470, and 491.
Point 13: Minor concerns: l. 271, 299, 379 - double In, there are text formatting issues complicating its assessment (l. 299-300, 316-340, 354-369), dried in an oven (l. 222), β-ionone ring (l. 288).
Reply: We appreciate your meticulous review of our manuscript. We have thoroughly addressed the formatting issues highlighted.
Point 14: In general, English is good, but there are some minor mistakes (e.g. in the first step should be at the first step, In the second step, missing point before respectively, etc).
Reply: Thank you for your feedback. We have revised the manuscript to avoid any minor mistakes.
Thank you once again for your valuable insights and comments, which are truly helping us improve our work.
Round 3
Reviewer 1 Report
I am grateful to the authors for their detailed responses. Now the text is ready for publication.
*Note that the absence of astaxanthin esters in your datasets could be due to low retention times in your HPLC protocol. Due to their high hydrophobicity, they are eluted after free carotenoids, excluding carotenes.
Minor editing of English language required. I belive, it will be done at the stage of English correction.